# Effect of Annealing Treatment on Sensing Response of Inorganic Film Taste Sensor to Sweet Substances

**DOI:** 10.3390/s25061859

**Published:** 2025-03-17

**Authors:** Tomoki Shinta, Hidekazu Uchida, Yuki Hasegawa

**Affiliations:** Graduate School of Science and Engineering, Saitama University, Shimo-Okubo 255, Sakura-ku, Saitama 338-8570, Japan; hiuchida@mail.saitama-u.ac.jp

**Keywords:** taste sensor, annealing treatment, XRD, SEM, inorganic film, sputter deposition, SnO_2_, sweet substances

## Abstract

The effect of annealing treatment on an inorganic film for taste sensors has not been fully elucidated. In this study, we developed an inorganic film taste sensor using SnO_2_ as a sensitive film and evaluated the effect of annealing treatment on its sensing response to sweet substances. First, we confirmed from XRD patterns that annealing at 600 °C caused a change in crystal orientation. Next, the taste sensor response to acesulfame potassium solution, which is a high-intensity sweetener and an electrolyte, showed a negative response with high concentration dependence. On the other hand, the sensors exhibited a positive response to non-electrolytes such as aspartame and glucose, with the sensor annealed at 600 °C showing a larger response to non-electrolytes compared to the other sensors. In terms of concentration dependence, the response to aspartame was higher, whereas the response to glucose was lower. Also, a reduction in variability was observed after annealing treatment at 150 °C and 300 °C. This phenomenon was clarified by comprehensively investigating various properties.

## 1. Introduction

In general, beverage manufacturers and the food service industry use sensory tests by trained inspectors to evaluate the taste of beverages in various situations, such as new product development and product quality control. However, sensory testing requires a lot of time and money to train inspectors, and is affected by the inspector’s preferences and physical condition at the time of the test, resulting in problems such as a lack of objectivity and reproducibility. Therefore, it is necessary to establish an objective evaluation method. Testing beverage quality is evaluated using optical analysis methods such as spectroscopy and fluorescence, and chromatography methods that separate substances into their components [1,2,3,4,5,6,7]. These methods can identify the type and amount of a substance, but cannot evaluate actual tastes such as sweetness and bitterness. One of the commercially available taste sensors that can evaluate the taste itself uses an artificial lipid membrane as the sensing membrane that mimics the mechanism of biological taste receptors. It has been reported that lipid membrane taste sensors can quantify the five basic tastes (sourness, saltiness, umami, bitterness, and sweetness) and astringency [8,9,10,11,12,13,14]. The lipid membrane taste sensors have problems with stability and durability when used with hot beverages because the lipid membranes used as the sensitive membranes of taste sensors undergo a phase transition at temperatures of 45 °C or higher [15,16,17]. Therefore, we are attempting to develop taste sensors using some inorganic films [18], which is generally used as the sensitive film of gas sensors and can withstand high temperatures [19,20,21,22]. It has been shown that inorganic film taste sensors can measure hot beverages and are suitable for measuring electrolytes [18]. The sensors have also been shown to have the potential to capture interactions that occur in human taste sensations. However, the response mechanism of inorganic film taste sensors has not been elucidated, and there are issues with sensitivity and stability. Further research is needed to improve the sensitivity and stability of the sensor system.

In general, annealing treatment is used in gas sensors as a method to improve their sensitivity and stability [19,20]. Several studies have reported that annealing treatment improves the crystallinity of sensitive films such as tin(IV) oxide (SnO_2_) and titanium(IV) oxide (TiO_2_), and changes the crystal orientation and surface structure of the sensor [21,22,23,24,25]. In addition, it has been revealed that changes in the crystallinity and surface structure of the sensitive film affect the sensitivity and responsiveness of the sensor [19,26]. SnO_2_ is a mainstream material for the sensitive film of gas sensors, and many studies have been conducted on it. In one of the papers about SnO_2_ by Mehraj, S. et al., it was confirmed that when the annealing temperatures were in the range from 600 °C to 900 °C, the crystallinity was improved using X-ray diffraction (XRD), but when the annealing temperature was 800 °C or lower, no significant changes in the surface structure were observed using a scanning electron microscope (SEM) [26]. The relationship between the properties of the inorganic film and the response of the taste sensor has not been clarified.

The purpose of this study is to clarify the effect of annealing treatment on inorganic film taste sensors and contribute to elucidating their response mechanism. Therefore, we developed a taste sensor using SnO_2_ as an inorganic sensitive film by RF magnetron sputtering and annealed it at 150, 300, 450, and 600 °C. We evaluated the differences in the sensor response to sweet substances such as sugar (glucose) and high-intensity sweeteners (acesulfame potassium, aspartame). By elucidating the response mechanism of the inorganic film taste sensor, we expect to establish a sensor system that can be widely applied to hot beverages.

## 2. Materials and Methods

### 2.1. Fabrication and Property Measurement Method of Inorganic Film Taste Sensors

The structure of the sensitive film of our taste sensor is shown in Figure 1. First, an alumina substrate (5 mm × 15 mm × 0.38 mm, Japan Fine Ceramics Co., Ltd., Miyagi, Japan) was ultra-sonically cleaned sequentially with acetone, methanol, and distilled water, for 10 min in each solvent. Silver (Ag) and platinum (Pt) layers were formed as an electrode, and a SnO_2_ layer was formed as a sensitive film, both with dimensions of 5 mm × 10 mm. The thicknesses of Pt, Ag, and SnO_2_ layers were 100 nm, 200 nm, and 200 nm, respectively. Each layer was deposited using the RF magnetron sputtering method (Canon Anelva Co., Ltd., Kanagawa, Japan: SPF-210H) under the conditions shown in Table 1. Next, a gold (Au) layer was coated to prevent the Ag layer from oxidizing during the annealing process (Figure 1a). Annealing treatment was performed in an oxygen atmosphere (flow rate: 10 cm^3^/min) for 60 min at 150, 300, 450, and 600 °C using a tubular electric furnace. Finally, the electrode was soldered to a printed circuit board, and everything except the sensitive film was insulated with epoxy resin (Figure 1b). The surface morphology of the taste sensor fabricated in this experiment was measured by an SEM (Hitachi High-Tech Co., Ltd., Tokyo, Japan: S-4800), and the crystallinity was measured by an XRD device (Bruker Co., Ltd., Billerica, MA, USA: D8 DISCOVER).

### 2.2. Measurement Method of Taste Sensor

A schematic diagram of the taste sensor measurement system is shown in Figure 2. The measurement cell contained 100 mL of 1 mM potassium chloride (KCl, Fujifilm Wako Pure Chemical Co., Ltd., Osaka, Japan) aqueous solution at 23 °C as the reference solution, which was stirred, and the developed taste sensor was immersed in the solution. The reference cell contained saturated KCl aqueous solution and a reference electrode (Ag/AgCl, Thermo Fisher Scientific Inc., Waltham, MA, USA; 900200) was immersed in the solution. The measurement and reference cells were electrically connected via a salt bridge prepared by dissolving saturated KCl in agar (C_15_H_31_COOH, Fujifilm Wako Pure Chemical Co., Ltd., Osaka, Japan).

The electric potential difference between the taste sensor and the reference electrode was measured using a potentiometer (Graphtec Co., Ltd., Kanagawa, Japan: GL840-SDM) with a sampling interval of 250 ms. The response value (in Figure 3) was calculated as the difference between the average values 10 s before and 5 s after the drop time of the 10 mL sample solution. All measurements were performed 10 times for each taste substance and concentration, and the average response values with standard errors are presented in the results.

### 2.3. Sample Solutions

The measurement samples were obtained from Fujifilm Wako Pure Chemical Co., Ltd., Osaka, Japan. All samples were diluted in a 1 mM KCl aqueous solution. The high-intensity sweeteners, acesulfame potassium (C_4_H_4_KNO_4_S) and aspartame (C_14_H_18_N_2_O_5_), were tested at concentrations of 0.0091, 0.045, 0.091, 0.45, and 0.91 mM. Acesulfame potassium is a potassium salt containing a cyclic oxathiazine dioxide group and exhibits low hydrophobicity. When dissolved in water, it ionizes into potassium cations and acesulfame anions, making it a water-soluble electrolyte. Aspartame is a dipeptide composed of L-aspartic acid and L-phenylalanine, linked by a peptide bond with a terminal methyl ester group. It is classified as a non-electrolyte because it does not ionize in water. For the sugar, D(+)-glucose (C_6_H_12_O_6_) was tested at concentrations of 0.91, 4.5, 9.1, 45, and 91 mM. D(+)-glucose exists in several isomeric forms that interconvert between linear and cyclic structures and is a non-electrolyte in water. The properties of sample solutions are shown in Table 2. Notably, the concentration range of each sample solution was set based on the human taste threshold [27,28,29].

## 3. Results

### 3.1. Physical Properties (SEM, XRD)

Figure 4 shows SEM images of the SnO_2_ layer of the taste sensor in its as-deposited state (Figure 4a) and after annealing at different temperatures (Figure 4b–e). In all cases, grain sizes of approximately 100 nm were observed across the entire surface. It was confirmed that the annealing treatment did not significantly change the surface structure, consistent with findings of a previous study [26]. Figure 5 shows the measurement results obtained through XRD using the grazing-incidence X-ray diffraction (GIXRD) scan mode on the SnO_2_ layers of the same devices shown in Figure 4a–e. The diffraction peak of SnO_2_ (200) was observed in the sensors as-deposited and after annealing at 150 to 450 °C, while the diffraction peak of SnO_2_ (111) appeared in the sensor annealed at 600 °C. The change in the crystal plane was observed only after annealing at 600 °C, and the change in the crystal structure caused by annealing was consistent with the results of previous studies [26,30].

### 3.2. Sensor Response

First, we show the results of taste sensor responses to high-intensity sweeteners. The responses to acesulfame potassium solution are shown in Figure 6. Negative responses were observed, and concentration dependences were confirmed for all sensors. It also shows that the response to acesulfame potassium solution does not change with annealing temperatures from 150 °C to 600 °C, and compared to the as-deposited sensor, the annealed sensors had smaller response values. It was observed that the error range was smaller for the as-deposited and annealed states at 300 °C. The calibration curve of the as-deposited state is also shown in Figure 6. In addition, the slopes of the calibration curves for all sample solutions are shown in Table 3. For sensors that did not show a concentration dependency, evaluation using a calibration curve was not performed (indicated with “-”in Table 3). In Table 3, the annealed sensor was observed to have a smaller slope of the calibration curve to acesulfame potassium.

The responses of taste sensors to aspartame solution are shown in Figure 7. When the sample solution was dropped, positive responses were observed. The as-deposited, 150 °C, 300 °C, and 600 °C sensors exhibited concentration dependence with small errors observed at high concentrations. In contrast, the response value of the sensor annealed at 450 °C decreased at 0.091 mM. It was also confirmed that the sensor annealed at 600 °C showed the largest response values and a steeper slope in the calibration curve (Table 3).

Next, the responses to taste sensors to glucose, one of the common sugars, are shown in Figure 8. The dispersion of measurement values was large for all sensors, and no concentration dependence was observed.

When sweet substances were dropped into the measurement cell, only the acesulfame potassium solution, which is an electrolyte, showed a negative response, whereas non-electrolytes such as aspartame and glucose exhibited positive responses. The dependency properties differed between aspartame and glucose.

### 3.3. Variability in Sensor Responses

In Section 3.2, it was confirmed that the responses of the taste sensors to some sample solutions and their concentrations had large error ranges (there were 10 trial times). Therefore, we evaluated the variability in sensor responses to each sample solution in the middle of the measured concentration ranges. Figure 9 shows the response of the taste sensors when 0.091 mM acesulfame potassium solution was dropped. For the as-deposited state, the sensor responses gradually increased from −2 mV to −4 mV. At 150 °C and 300 °C, stable variability was confirmed, and variations of ±0.5 mV and ±0.75 mV, respectively, were observed around −1 mV. At 450 °C and 600 °C, some outlier positive responses were observed.

Then, the response results for a 0.091 mM aspartame solution are shown in Figure 10. In the as-deposited, 150 °C, and 300 °C states, a variation of ±1.7 mV was observed around 2 mV. At 450 °C, several negative outliers were observed. It was confirmed that the measurements were unstable at 600 °C.

The response of the taste sensor with 9.1 mM glucose solution is shown in Figure 11. In the as deposited, 150 °C, 300 °C, and 450 °C states, a variation of ±2.0 mV was observed around 1.5 mV. An outlier was observed at 600 °C.

The annealing temperature of the taste sensor showed a tendency for the error bars to vary. The error bars were larger at higher annealing temperatures, and tended to be smaller at 150 °C and 300 °C. In order to reduce the error range by this result, it is necessary to consider conditions such as annealing temperature and time.

## 4. Discussion

In general, annealing treatment is used in gas sensors to improve their sensitivity and stability. It enhances the crystallinity of sensitive films such as SnO_2_ and changes the crystal orientation and surface structure of the sensor. Additionally, it has been revealed that changes in the crystallinity and surface structure of the sensitive film affect the sensitivity and responsiveness of the sensor.

The taste sensor response to the acesulfame potassium solution, which is an electrolyte, showed a negative response. In a previous study, the sensor also showed a negative response to NaCl solution [18]. These results indicate that the sensor using SnO_2_ as a sensitive film exhibits a negative response to ions such as Na^+^ and K^+^. When aspartame solution was dropped, the sensor annealed at 600 °C showed a larger response value compared with other sensors. In Figure 4, no significant change was observed in the surface structure of the sensor when annealed at temperatures below 600 °C. In contrast, a difference in crystallinity was observed in the sensor annealed at 600 °C from the XRD measurement results in Figure 5. From these results, it is thought that the response to non-electrolytes or hydrophobic substances changes due to differences in crystallinity. Aspartame has both hydrophilic and hydrophobic properties, whereas glucose has only hydrophilic properties. The concentration dependencies differed between the non-electrolytes aspartame and glucose. However, the reason for this difference remains unclear and requires further investigation.

The variability in sensor responses was examined and the measurement variability was within 2 mV at deposition, 150 °C, and 300 °C. Additionally, outliers were observed at 450 °C and 600 °C when the crystal structure changed. This difference in variation is thought to be due to the effect of the difference in annealing temperature on the interface. The physical properties of the interface between the inorganic film and the electrode also require further investigation.

## 5. Conclusions

In this study, we evaluated the difference in the response of an inorganic film taste sensor by changing its physical properties through annealing treatment. Taste sensors using SnO_2_ as an inorganic sensitive film were fabricated by RF magnetron sputtering, annealed at temperatures of 150, 300, 450, and 600 °C, and compared with the as-deposited sensor. The differences in the sensor response to sweetness (acesulfame potassium, aspartame, and glucose) were evaluated to confirm the effect of annealing on the inorganic film taste sensor. When the inorganic film taste sensor was annealed at 600 °C, changes in the crystal plane were observed. The inorganic film taste sensors indicated that annealing treatment at temperatures below 600 °C does not change the response to the electrolyte (acesulfame potassium). However, it was confirmed that the response to hydrophobic substances, aspartame, changed upon annealing. Also, no concentration dependence was observed for any of the sensors for glucose, which is a non-electrolyte. Additionally, it was confirmed that the sensors annealed at 150 °C and 300 °C had small errors. In the future, we will comprehensively investigate and clarify the response of the taste substances in aqueous solution and the changes in the sensor due to annealing treatment.

This result suggests that annealing treatment can be used to develop a highly sensitive sensor for hydrophobic substances. We attempt to further elucidate the response mechanism by evaluating different physical properties using techniques such as Transmission Electron Microscopy (TEM) and X-ray Photoelectron Spectroscopy (XPS), as well as XRD, and by changing the crystal structure and the material exposed on the crystal surface under different processing and deposition methods.

## Figures and Tables

**Figure 1 sensors-25-01859-f001:**
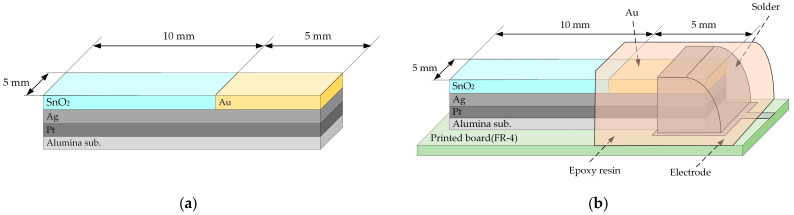
Structures of the taste sensor. (**a**) Pt, Ag, SnO_2_, and Au layers were deposited on an alumina substrate by RF magnetron sputtering; (**b**) the sensor device was soldered onto the printed circuit board and covered by epoxy resin except for the sensing area.

**Figure 2 sensors-25-01859-f002:**
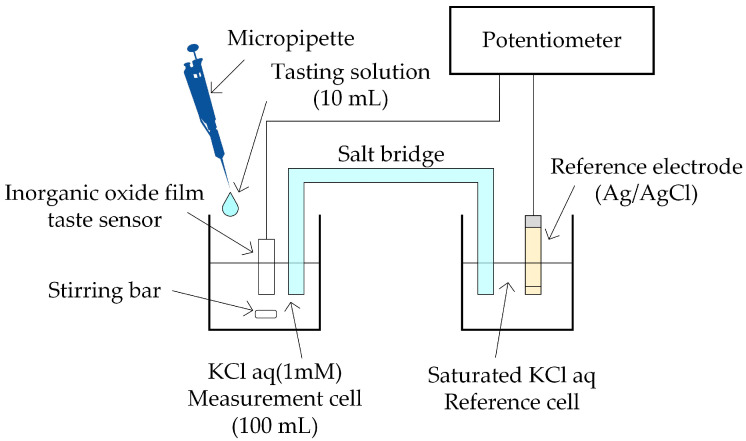
Schematic diagram of the taste sensor measurement system.

**Figure 3 sensors-25-01859-f003:**
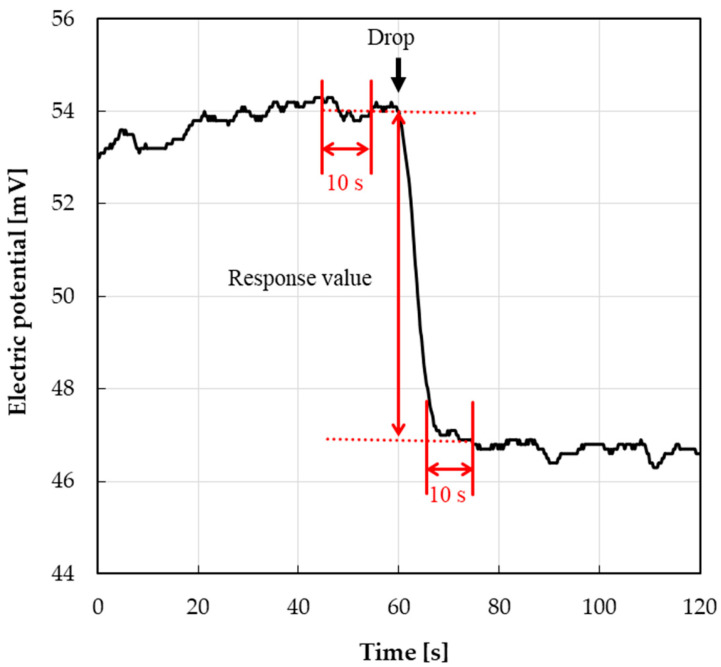
Typical sensor response to dropped solution and the definition of the response value.

**Figure 4 sensors-25-01859-f004:**
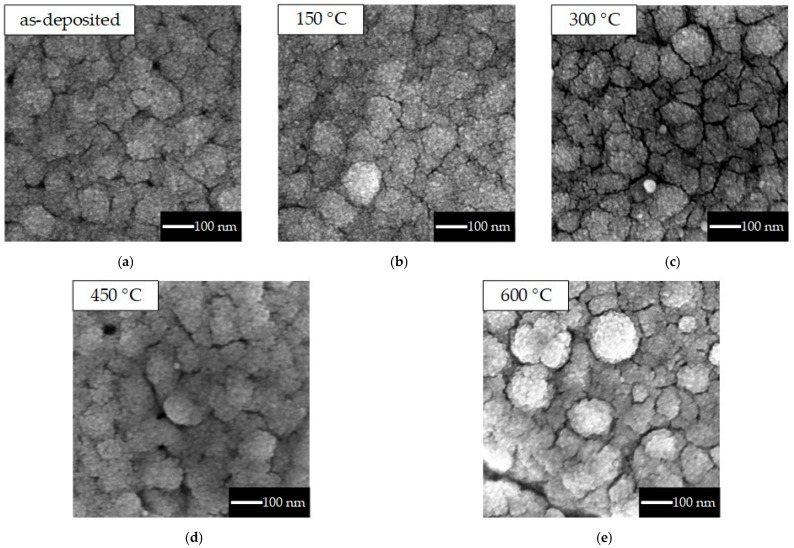
SEM images of as-deposited and annealed SnO_2_ layer of taste sensor: (**a**) as-deposited, (**b**) 150 °C, (**c**) 300 °C, (**d**) 450 °C, and (**e**) 600 °C.

**Figure 5 sensors-25-01859-f005:**
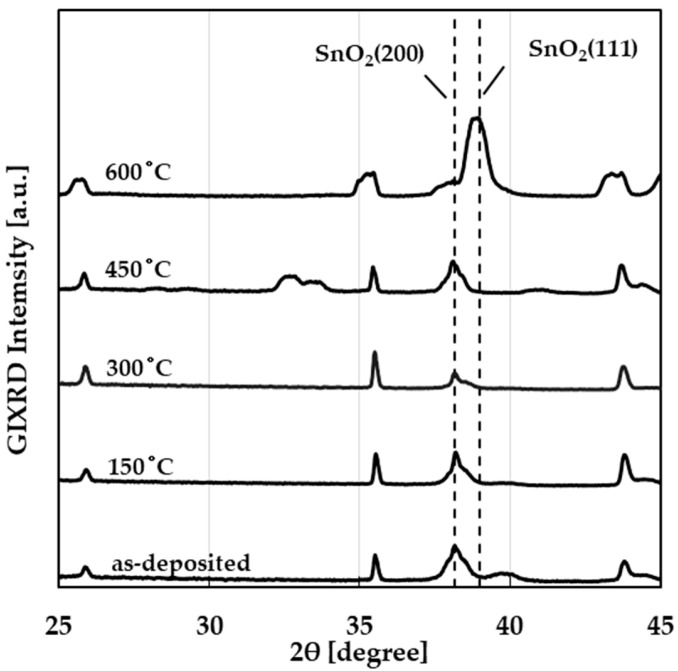
XRD patterns from GIXRD scan of as-deposited and annealed (150, 300, 450, and 600 °C) SnO_2_ layer of taste sensors.

**Figure 6 sensors-25-01859-f006:**
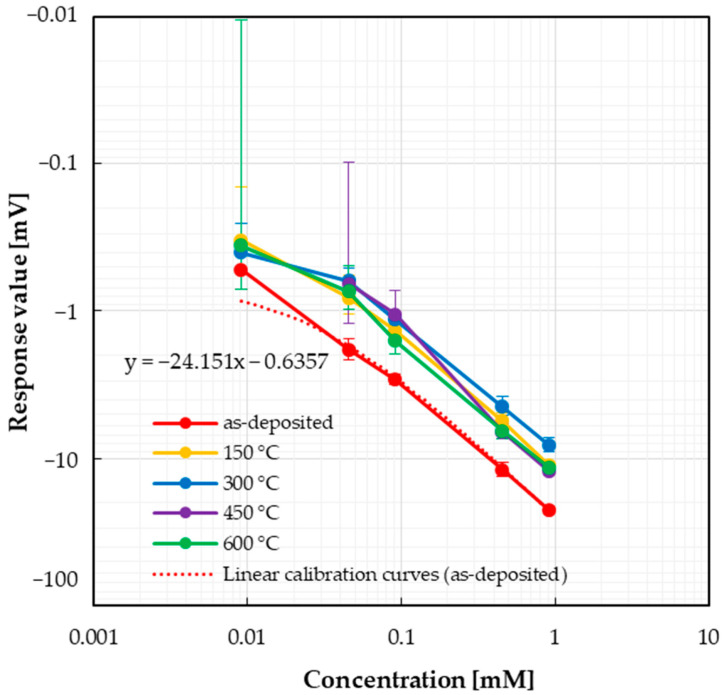
Responses of taste sensor to acesulfame potassium.

**Figure 7 sensors-25-01859-f007:**
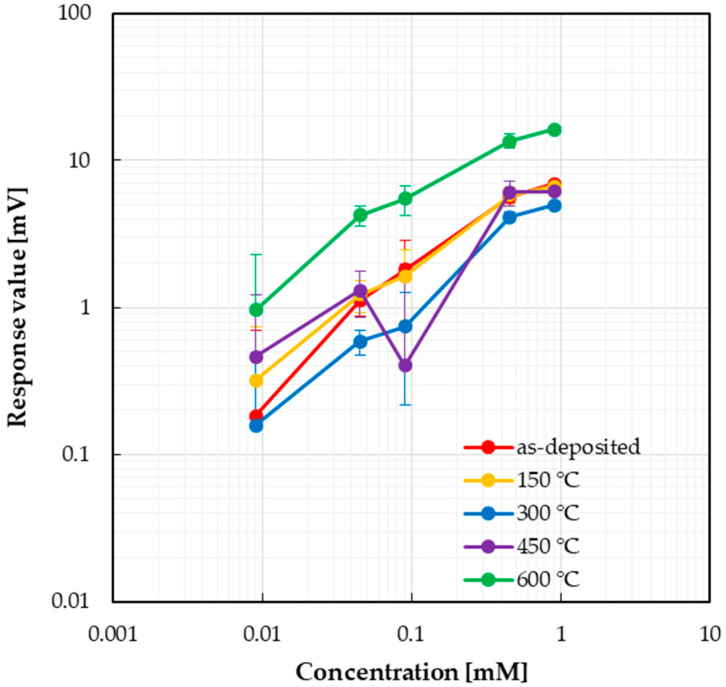
Responses of taste sensor to aspartame.

**Figure 8 sensors-25-01859-f008:**
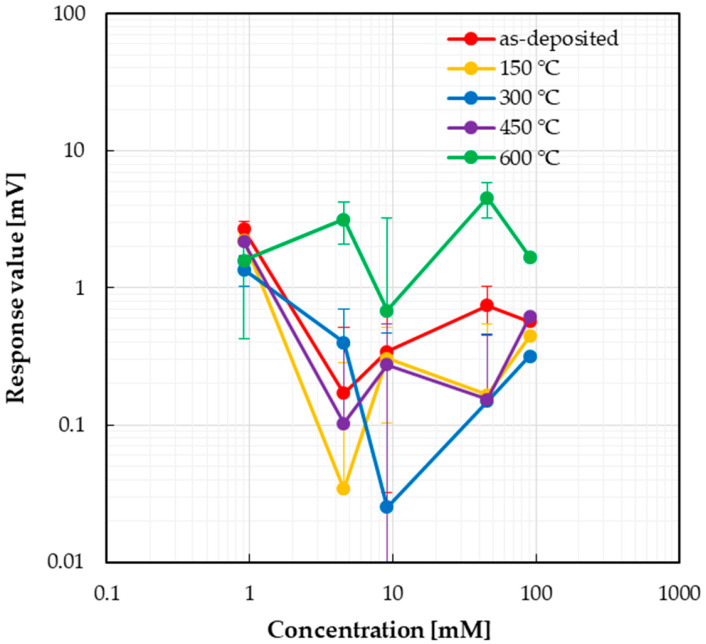
Responses of taste sensor to glucose.

**Figure 9 sensors-25-01859-f009:**
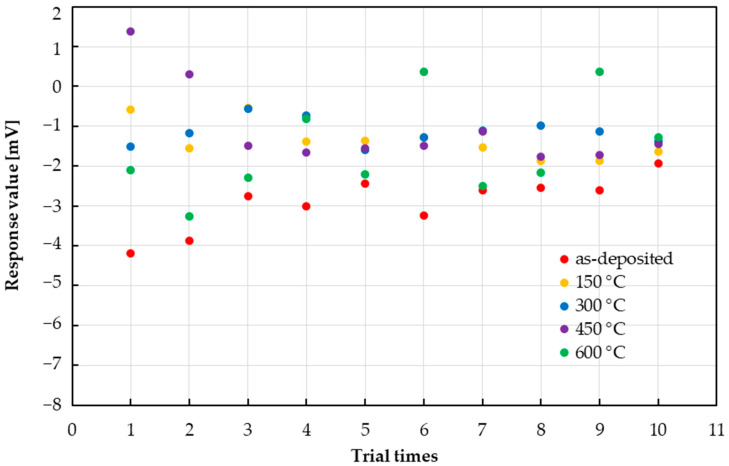
Variability in responses of taste sensors to acesulfame potassium when 0.091 mM is dropped.

**Figure 10 sensors-25-01859-f010:**
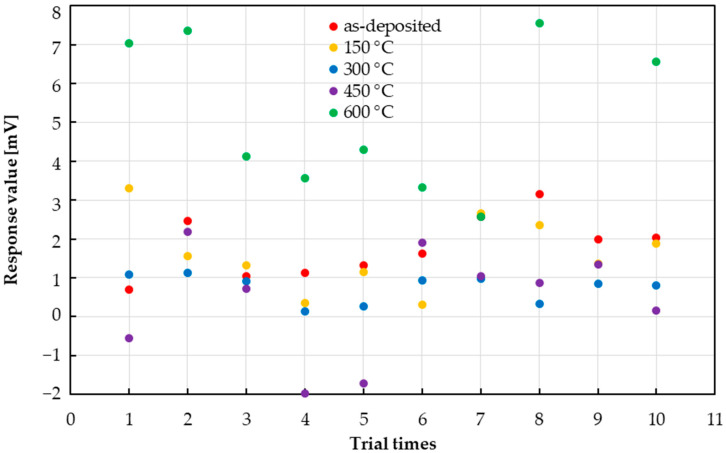
Variability in responses of taste sensors to aspartame when 0.091 mM is dropped.

**Figure 11 sensors-25-01859-f011:**
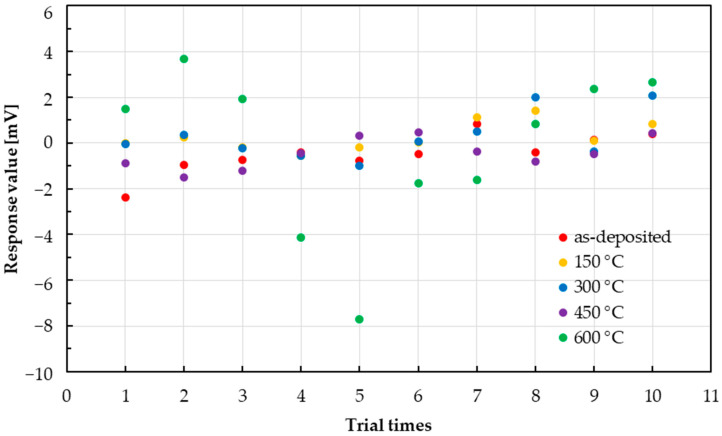
Variability in responses of taste sensor to glucose when 9.1 mM is dropped.

**Table 1 sensors-25-01859-t001:** Deposition conditions of thin films by RF magnetron sputtering.

Parameters	Conditions
Target	Pt, Ag, Au, SnO_2_ (4N)
Substrate	Alumina (3N)
Deposited temperature	R.T.
Deposited time	5 min
Sputtering power	50 W
Source gas	Ar (6N)
Flow rate	0.5 mL/m

**Table 2 sensors-25-01859-t002:** The properties of sample solutions.

Sample (Taste Threshold)	Chemical Formula (Molecular Weight)	Properties	In Water
Acesulfame potassium (0.5 mM) [27,28]	C_4_H_4_KNO_4_S (201.24)	Water-soluble, electrolytes, low hydrophobicity	K^+^ + C_4_H_4_NO_4_S^−^
Aspartame (0.5 mM) [27,28]	C_14_H_18_N_2_O_5_ (294.30)	Water-soluble, non-electrolytes, hydrophilic–hydrophobic	C_14_H_18_N_2_O_5_
D(+)-glucose (100 mM) [29]	C_6_H_12_O_6_ (180.16)	Water-soluble, non-electrolytes	C_6_H_12_O_6_

**Table 3 sensors-25-01859-t003:** The slopes of the fitted curve between the sample solutions and the taste sensors.

Sample	As-Deposited	150 °C	300 °C	450 °C	600 °C
Acesulfame potassium	−24.151	−11.941	−8.645	−13.576	−12.499
Aspartame	7.3915	7.128	5.561	-	16.002
D(+)-glucose	-	-	-	-	-

## Data Availability

The datasets generated for this study are available on request to the corresponding author.

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
