# Peer review of "Effect of Annealing Treatment on Sensing Response of Inorganic Film Taste Sensor to Sweet Substances"

_sensors, 2025, doi:10.3390/s25061859_

Round 1
Reviewer 1 Report
Comments and Suggestions for Authors
The manuscript entitled " Effect of Annealing Treatment on Sensing Response of Inor- 2ganic Film Taste Sensor ",in this research, authers fabricated an inorganic thin-film taste sensor utilizing SnO2 as the sensitive layer and investigated how annealing treatment impacts its sensory response to sweet and bitter compounds. By elucidating the response mechanism of the inorganic film taste sensor, it is expected to establish a sensor system that can be widely applied to hot beverages. The author's study is very interesting, but there are some key issues in the paper that need to be addressed.
1. In reviewing this manuscript, I noticed that the layout and design of the composite figures in the article have some issues, which affect the overall aesthetics and professionalism of the article. Specifically, the unbalanced layout of Figure 3 not only impacts the reader's experience but also reduces the overall aesthetic appeal of the article.
2. As is well known, the successful construction of a sensor is verified by the linear relationship between the concentration of the analyte and the detection signal, as well as LOD. However, the taste sensor constructed by the authors shows a linear relationship only for the response to acesulfame potassium. In the detection of aspartame, glucose, and quinine hydrochloride, the results do not conform to a linear relationship. Moreover, the authors did not mention LOD for any of the tested substances.
3. In Figure 5, Figure 6, Figure 7, and Figure 8, the authors used error bars to represent data stability. However, the error bars in these figures have such large errors that they compromise the credibility of the data.
4. The introduction section contains an excessive number of paragraphs, which makes the overall structure appear lengthy and complex. The abundance of paragraphs may cause readers to lose focus during the reading process, making it difficult to quickly grasp the core content and research background of the article.
5. In the mechanistic explanation, the author attributes the changes in the sensor's detection performance after calcination to changes in crystallinity, which seems somewhat speculative when explained solely through XRD. It would be beneficial to supplement the material with TEM to further validate these changes.
6. In the conclusion section, the author should briefly summarize the experimental results. Currently, the number of paragraphs in the results section is excessive, which fails to highlight the main contributions and innovations of the study.
Author Response
Thank you very much for reviewing my paper.
First, we would like to revise the descriptions of our measurement concentrations in our submitted paper, we are so sorry.
Initially, we had described the concentration of the dropped solutions to the cell as the measurement concentration. However, the concentration should have been shown as the concentration in the measurement cell. Therefore, the relevant parts of Section 2.3. Sample solutions, horizontal axis of Figure. 6, 7, 8, and 3. Results have been revised.
Second, as other reviewer suggested that “Need clear and objective introduction,” we focused on measuring sweet substance and made the purpose clear in this paper. In this time, we deleted the relevant parts of the descriptions of quinine hydrochloride. We would like to discuss the sensor responses to bitterness and other substances in our next paper. Also, if we could change our paper’s title and keywords, we would like to add "sweet substances".
Third, some reviewers commented about the stability and the variability of the sensors. So, we added the new section “3.3. Variability in sensor responses” to 3. Results and a new paragraph to 4. Discussions due to describe that point deeply.
Our responses to your comments are as follows, and we have highlighted in red the words or sentences that we had corrected or added on our revised paper:
Comments 1: In reviewing this manuscript, I noticed that the layout and design of the composite figures in the article have some issues, which affect the overall aesthetics and professionalism of the article. Specifically, the unbalanced layout of Figure 3 not only impacts on the reader's experience but also reduces the overall aesthetic appeal of the article.
Response 1: The font and character size have been changed for the figures.
Individual changes are as follows.
・Fig1: Font changed.
・Fig2: Font changed and the magnetic stirrer was removed as it looked unnatural.
・Fig3: Font and character size changed. “Before response value” and “After response value” in the figure have been deleted.
・Fig4: Font changed.
・Fig5: Font and character size changed.
・Fig6: Font and character size changed. Graph frame made thicker.
・Fig7: Font and character size changed. Graph frame made thicker. This was the only graph that was not standardized as a square, so the aspect ratio was corrected.
・Fig8: Font and character size changed. Graph frame made thicker.
Comments 2: As is well known, the successful construction of a sensor is verified by the linear relationship between the concentration of the analyte and the detection signal, as well as LOD. However, the taste sensor constructed by the authors shows a linear relationship only for the response to acesulfame potassium. In the detection of aspartame, glucose, and quinine hydrochloride, the results do not conform to a linear relationship. Moreover, the authors did not mention LOD for any of the tested substances.
Response 2: Thank you for your suggestion. We didn’t indicate LOD like general sensor’s research. Because our goal is to develop a taste sensor that evaluates the same taste with the human taste. For that reason, the measurement range is determined based on the taste threshold of each taste substance. We think that LODs for acesulfame potassium and aspartame in this study are at even lower concentrations. We added the human threshold of each taste substance in Table 2, and also added a new sentence about this explanation on page. 4, lines 131-132.
Comments 3: In Figure 5, Figure 6, Figure 7, and Figure 8, the authors used error bars to represent data stability. However, the error bars in these figures have such large errors that they compromise the credibility of the data.
Response 3: For all sensors, the error bars on the high concentration side were small, and the error bars on the low concentration side were large. For some sensors, the variability was improving depending on the annealing temperature. The results are added in 3.3. Variability of sensor response. To further improve the variability, we think it is necessary to find optimal annealing conditions.
Comments 4: The introduction section contains an excessive number of paragraphs, which makes the overall structure appear lengthy and complex. The abundance of paragraphs may cause readers to lose focus during the reading process, making it difficult to quickly grasp the core content and research background of the article.
Response 4: We reduced paragraphs in the introduction section depending on your advice.
Comments 5: In the mechanistic explanation, the author attributes the changes in the sensor's detection performance after calcination to changes in crystallinity, which seems somewhat speculative when explained solely through XRD. It would be beneficial to supplement the material with TEM to further validate these changes.
Response 5: Thank you for your advice. In discussing crystalline, we plan to additionally confirm it using electron beam diffraction images through TEM and fluorescent X-rays using XPS. We also thought that TEM can be used to investigate differences in surface images with SEM. We have added these prospects to "5. Conclusions" under other physical property evaluations.
Comments 6: In the conclusion section, the author should briefly summarize the experimental results. Currently, the number of paragraphs in the results section is excessive, which fails to highlight the main contributions and innovations of the study.
Response 6: We added some missing summaries. And we divided paragraphs into the part of results and the part of contributions and innovations. Also, we have added the innovative aspects of this study to the conclusions.
Thank you.
Reviewer 2 Report
Comments and Suggestions for Authors
Need clear and objective introduction.
A suggestion is a more comprehensive investigation into the sensor response mechanisms. In addition, a comparison with other sensitive materials would be interesting to better contextualize the results obtained.
There is a lack of a better relationship as to why the crystallinity of the film annealed at 600 degrees presents better sensitivity.
Need a clearer and more direct explanation of why the interaction occurs with aspartame, acesulfame potassium and quinine hydrochloride is positive and with ions such as Na⁺ and K⁺ and other electrolytes is negative.
Need a clearer and more direct explanation of why the interaction occurs with aspartame, acesulfame potassium and quinine hydrochloride is positive.
Author Response
Thank you very much for reviewing my paper.
First, we would like to revise the descriptions of our measurement concentrations in our submitted paper, I’m so sorry.
Initially, we had described the concentration of the dropped solutions to the cell as the measurement concentration. However, the concentration should have been shown as the concentration in the measurement cell. Therefore, the relevant parts of Section 2.3. Sample solutions, horizontal axis of Figure. 6, 7, 8, and 3. Results have been revised.
Second, your first comment suggested that “Need clear and objective introduction,” so we focused on measuring sweet substance and made the purpose clear in this paper. In this time, we deleted the relevant parts of the descriptions of quinine hydrochloride. We would like to discuss the sensor responses to bitterness and other substances in our next paper. Also, if we could change our paper’s title and keywords, we would like to add "sweet substances".
Third, some reviewers commented about the stability and the variability of the sensors. So, we added the new section “3.3. Variability in sensor responses” to 3. Results and a new paragraph to 4. Discussions due to describe that point deeply.
Our responses to your comments are as follows:
Comments 1: Need clear and objective introduction.
Response 1: We made the changes as stated above.
Comments 2: A suggestion is a more comprehensive investigation into the sensor response mechanisms. In addition, a comparison with other sensitive materials would be interesting to better contextualize the results obtained.
There is a lack of a better relationship as to why the crystallinity of the film annealed at 600 degrees presents better sensitivity.
Response 2: Thank you for your advice. In this paper, we used only SnO2 as a sensitive film of the taste sensor. As you know, SnO2 is often used as a sensitive film of gas sensors. Because of the variety of information available, we selected SnO2 as the sensitive film of our taste sensor. In gas sensors, a relationship between high crystallinity and high sensitivity has been shown. We expect that there will also be a relationship between crystallinity and sensitivity in taste sensors. We must investigate other sensitive materials, such as WO3, ZnO, In2O3, and so on, in near future.
Also, it is not yet clear whether crystalline is the cause. However, we think that the increase in the response value is due to the reduction in the resistance of the interface between the inorganic film and the electrode caused by the annealing treatment. We will continue to investigate this. As we mentioned briefly in "5. Conclusions", We plan to change the material and conduct experiments, and to clarify the response mechanism in the future.
Comments 3: Need a clearer and more direct explanation of why the interaction occurs with aspartame, acesulfame potassium and quinine hydrochloride is positive and with ions such as Na⁺ and K⁺ and other electrolytes is negative. Need a clearer and more direct explanation of why the interaction occurs with aspartame, acesulfame potassium and quinine hydrochloride is positive.
Response 3: Our experiments used a few ionic substances, so it is difficult to discuss the point now. However, we previously used sodium chloride to evaluate the difference in taste. We think that it is necessary to evaluate various ionic substances in the future. We have briefed it to the discussion.
Thank you.
Reviewer 3 Report
Comments and Suggestions for Authors
1. (Page 2, Line 61) The reviewer kindly recommends that the authors clarify the statement regarding temperature specifics. For instance, it would be helpful to specify ‘600 oC or higher’ and ‘800 oC or lower’.
2. (Page 2, Line 79) Could the authors elaborate on the function of the silver (Ag) layer within the taste sensor? Additionally, is the sensor functional with only a platinum deposition without the Ag layer?
3. (Table 1) Please provide the thickness of each constituent layer (Pt, Ag, Au, SnO2), as well as the total thickness of the electrodes.
4. (Page 6, Line 170) Could the authors please provide some description or explanation for why the response value of the sensor annealed at 450 oC decreases at a 1mM concentration of both acesulfame potassium and aspartame?
5. (Page 7, Line 187) How were the calibration curves and the slope of the curves (0.214 and 0.426) derived?
6. (Minor comments)
- Figure 1(a). ‘SnO2’ should be corrected to ‘SnO2’
- No definition for the abbreviation ‘GIXRD’
Author Response
Thank you very much for reviewing my paper.
First, we would like to revise the descriptions about our measurement concentrations in our submitted paper, I’m so sorry.
Initially, we had described the concentration of the dropped solutions to the cell as the measurement concentration. However, the concentration should have been shown as the concentration in the measurement cell. Therefore, the relevant parts of Section 2.3. Sample solutions, horizontal axis of Figure. 6, 7, 8, and 3. Results have been revised.
Second, as other reviewer suggested that “Need clear and objective introduction,” we focused on measuring sweet substance and made the purpose clear in this paper. In this time, we deleted the relevant parts of the descriptions of quinine hydrochloride. We would like to discuss the sensor responses to bitterness and other substances in our next paper. Also, if we could change our paper’s title and keywords, we would like to add "sweet substances".
Third, some reviewers commented about the stability and the variability of the sensors. So, we added the new section “3.3. Variability in sensor responses” to 3. Results and a new paragraph to 4. Discussions due to describe that point deeply.
Our responses to your comments are as following:
Comments 1: (Page 2, Line 61) The reviewer kindly recommends that the authors clarify the statement regarding temperature specifics. For instance, it would be helpful to specify ‘600 °C or higher’ and ‘800 °C or lower’.
Response 1: The relevant part is based on "Alimuddin Annealed SnO2 Thin Films: Structural, Electrical and Their Magnetic Properties" written by Mehraj, S. This experiment was performed as-deposited, at 600-900 °C, and a change in the crystal structure was confirmed at 600 °C. Change this to "Temperature is from 600 °C to 900 °C."
Comments 2: (Page 2, Line 79) Could the authors elaborate on the function of the silver (Ag) layer within the taste sensor? Additionally, is the sensor functional with only a platinum deposition without the Ag layer?
Response 2: The silver layer is under the sensitive films, however the films is porous and have some small holes. Therefore, we considered the silver layer will contact with the reference solution, 1 mM potassium chloride. The Ag and Cl- react and become AgCl, then the initial potential of the sensor becomes stable. As a preliminary experiment, we tried to measure the sensor with only a platinum and confirmed the sensor response was unstable.
Comments 3: (Table 1) Please provide the thickness of each constituent layer (Pt, Ag, Au, SnO2), as well as the total thickness of the electrodes.
Response 3: Adding the film thicknesses to Table 1 would have made the layout of the figure unbalanced, so we decided to explain them in the text. Also, the thickness of the Au anti-oxidation film has not been measured, so it is not listed. The thicknesses were 100 nm for Pt, 200 nm for Ag, and 200 nm for SnO2.
Comments 4: (Page 6, Line 170) Could the authors please provide some description or explanation for why the response value of the sensor annealed at 450 ˚C decreases at a 1mM concentration of both acesulfame potassium and aspartame?
Response 4: Regarding 1 mM, we have determined that it indicates 0.091 mM, and will respond to you. We observed the response value of the sensor annealed at 450 ˚C decreases at a 0.091 mM concentration of only aspartame. We have no idea the reason why this result. From XRD result, a peak not seen anywhere else was observed at 450 ˚C at angles lower than 35° in Figure 5. We think this is one of the factors affecting the response value of the sensor at 450 ˚C. Details will be revealed in the future.
Comments 5: (Page 7, Line 187) How were the calibration curves and the slope of the curves (0.214 and 0.426) derived?
Response 5: The linear approximation equation was obtained using Microsoft Excel from the concentration dependence graph. The slope of -24.151 was obtained from the as-deposite plot in Figure 6. The results are summarized in Table 3.
Comments 6: (Minor comments)
- Figure 1(a). ‘SnO2’ should be corrected to ‘SnO2’
- No definition for the abbreviation ‘GIXRD’
Response 6: Thank you for your comment confirming the typo. Figure 1(a) has been revised. The definition for GIXRD has also been added.
Thank you.
Round 2
Reviewer 1 Report
Comments and Suggestions for Authors
In this study, the authors developed an inorganic thin-film taste sensor using SnO₂ as the sensitive membrane and evaluated the impact of annealing treatment on its response to sweet substances. The authors confirmed from the XRD patterns that annealing at 600°C induced changes in crystal orientation. Subsequently, the taste sensor exhibited a negative response to high-intensity sweeteners and the electrolyte potassium acesulfame solution, with a strong concentration dependence. In contrast, the sensor displayed a positive response to non-electrolytes, such as aspartame and glucose. The research topic is innovative, the experimental design is systematic, and the data presentation is comprehensive. This work provides a meaningful contribution to both practical applications and mechanistic studies in the field of sensors.
The authors have satisfactorily responded to all reviewer comments. The revised manuscript has been significantly improved and is now suitable for acceptance.